# Cost of Recommended Diet (CoRD) and Its Affordability in Bangladesh

**DOI:** 10.3390/foods12040790

**Published:** 2023-02-13

**Authors:** Saiful Islam, Abira Nowar, Md. Ruhul Amin, Nazma Shaheen

**Affiliations:** Institute of Nutrition and Food Science, University of Dhaka, Dhaka 1000, Bangladesh

**Keywords:** affordability, Bangladesh, cost of recommended diets, food-based dietary guidelines

## Abstract

The cost of diet has been recognized as a major determinant of overall diet quality and nutritional outcomes. We aimed to estimate the minimum cost and affordability of the recommended diet based on the updated food-based dietary guidelines (FBDG) in Bangladesh. To compute the cost of the recommended diet (CoRD), we collected retail prices of foods corresponding to each of the food groups in the latest Bangladeshi FBDG. For affordability, the household size and daily food expenditure data were used from the most recent Household Income and Expenditure survey (HIES). The CoRD was calculated based on the average number of servings recommended for each food group; the CoRD was adjusted by a deflation factor and divided by the household’s daily food expenditure to estimate affordability. We found that the CoRD was $0.87 (83 BDT) per person per day at the national level. Nationally, about 43% of households could not afford the CoRD, with rural areas bearing a disproportionate share of the burden. We also found households to overspend on starchy staples while underspending on protein-rich foods, fruits, and dairy. These findings highlight the need for immediate implementation of interventions to improve the affordability of the CoRD and redesign policy instruments to create a sustainable food system.

## 1. Introduction

The cost of recommended diets (CoRD) is an estimate of the basic minimum cost needed to follow the food-based dietary guidelines (FBDG). The cost of diet has increasingly been recognized as a major determinant of the quality of overall diets and nutrition outcomes. The relatively higher cost of healthy foods leads to reduced consumption of nutrients, resulting in compromised diets and micronutrient inadequacy. In fact, Dizon and Anna considered food cost as a major and presumably one of the limiting factors to accessing safe and healthy diets [1]. The latest report of the State of Food Security and Nutrition in the World (SOFI) estimated that globally 3 billion people are unable to afford a healthy diet [2]. Findings of the recent studies also highlighted that affording the cost of healthy and nutritious foods has remained an overarching goal for a significant portion of the world’s population, particularly for the poorest people [3,4,5]. Additionally, with the economic shock and instability caused by the COVID-19 pandemic, food prices have increased, making it more difficult to afford healthy diets, especially for people from low- and middle-income countries. As a result, achieving the Sustainable Development Goal 2 (SDG 2) targets of zero hunger, food security, and improved nutrition by 2030 is becoming a far-reaching dream [2].

Bangladesh, the world’s eighth most populous country, has made remarkable progress from its earlier estimates in economic growth, food production, health, and nutrition [6]. The country with a gross national per capita income of $1470 is scheduled to graduate from its current status to a developing country in 2026 [7]. The poverty rate has also appreciably dropped to 20.5% in 2019 from 24.3% in 2016 [8]. Despite all these gains, Bangladeshi diets continue to be dominated by rice, with less emphasis on non-cereals and a variety of other nutrient-rich foods. This scenario reflects the fact that 34% of infants and young children in Bangladesh have a minimum adequate diversity in their diets [9]. Latest estimates from food consumption surveys have shown that a diet composed of different foods in Bangladesh is far below in diet quality according to the definition of a “Healthy Diet” by World Health Organization (WHO) [10]. Moreover, the recent measures to mitigate the COVID-19 pandemic have further exacerbated household dietary quality in that 61% of families in Bangladesh reported consuming less diversified diets than their pre-pandemic diets [11].

To eradicate all forms of malnutrition and achieve food and nutrition security, there is a need to create a sustainable, resilient food system for healthy diets that meet the needs of the population in terms of energy and macro- and micro-nutrients. However, attaining food and nutrition security is not about just meeting energy and nutrient needs. It also entails the consumption of balanced and healthy diets, as promoted in the FBDG. FBDG are guidelines that are formed considering the dietary pattern, food habits, and culture of a population and incorporate recommendations that address major diet-related public health issues [12]. Such guidelines not only include the basic nutrient needs but go beyond and represent diets in a manner that provides overall health protection in order to eradicate all forms of malnutrition. The inability to afford sufficient, safe and nutritious foods is a critical driver of the lack of access to such recommended diets. As a result, it is critical to understand whether a country’s existing food systems can translate dietary guidelines into affordable consumption of recommended diets.

Existing literature on the cost and affordability of diets in Bangladesh is almost exclusively based on the Save the Children UK-developed cost-of-the-diet (CoD) methodology, which calculates the least cost of meeting essential nutrient requirements by typical households in a specific geographic location. In 2006, the CoD approach was first piloted in a village in the Rangpur district [13]. Subsequently, this analysis was conducted for the fish cultivation livelihood zone in Khulna [14] and beneficiary households of the Suchana program in the districts of Sylhet and Moulvibazar [15]. This methodology was also used by the World Food Programme in its Fill the Nutrient Gap (FNG) analyses in Bangladesh [16]. CoD or other similar analyses [17] are based on meeting only the bare minimum of critical nutrient requirements. However, they are useful for nutrition assistance/relief programs in planning diets for the poor or other specific vulnerable groups such as children [18]. In contrast to CoD, the Cost of Recommended Diet (CoRD) goes beyond essential nutrients to incorporate the cost of foods from different food groups as recommended in the FBDG. While the CoRD may suffer the limitation of not fully reflecting the food culture and consumption behavior of a particular population, this method is obviously superior to CoD when the goal is to promote overall protection and promotion of the health of the Bangladeshi population. This is because the CoRD is based on calculating the cost of meeting the FBDG recommendations, which include diversified foods with varying functionalities.

The method of calculating the cost of meeting recommendations in the FBDG, alias the Cost of Recommended Diet (CoRD), was pioneered in Ghana [19] and later applied in Africa [20], India [21], and Myanmar [22]. More recently, the CoRD has been assessed using a regional FBDG generated using national dietary guidelines of South Asian countries [1,23]. Although Bangladesh was included in the analysis, the CoRD and affordability estimates for Bangladesh were not truly based on Bangladesh-specific FBDG, which differs from other FBDGs from South Asian countries in terms of food groupings and amounts of foods in food groups. For example, Bangladeshi FBDG has nine food groups compared to only six in the regional (South Asian) FBDG. Also, the maximum recommended amount of vegetable and fruit groups are set at much higher, and that of the starchy staple at much lower amounts than the maximum recommended amount in the regional FBDG [1]. More importantly, the analysis relied on almost a decade-old FBDG [24] instead of the most recent one, revised and updated in 2020. Furthermore, it used households-reported price data collected back in 2014–2015 that do not fully reflect the food prices in the current context. Similarly, the recent report of the World Bank published in March 2021 also estimated the cost and affordability of a healthy diet in Bangladesh by following the outdated FBDG of 2013 [25]. With these considerations, this study aimed to generate the CoRD estimates using FBDG specifically developed for the Bangladeshi population (i.e., FBDG 2020) and retail food prices collected through a market survey. It also aimed to estimate the affordability of the CoRD with respect to household food expenditures from the Household Income and Expenditure Survey (HIES) of 2016 [26].

## 2. Methods

### 2.1. Food Selection

Figure 1 presents the overall methodology we followed. Calculating the CoRD required a list of food items that are frequently consumed and available in the food markets throughout Bangladesh. Foods listed in the Food Composition Table for Bangladesh (FCTB) were used as our initial food list, as it contains 386 Bangladeshi food items with their nutrient composition [27,28]. Later, to reflect the current consumption pattern, the food list was updated based on recent nutrition-related surveys, namely the Bangladesh Integrated Household Survey, 2015 (BIHS, 2015), Institute of Nutrition and Food Science Survey, 2017–2018 (INFS, 2017–2018), and Household Income and Expenditure Survey, 2016 (HIES, 2016). Subsequently, a comprehensive food list containing 124 food items was finalized. The food items were then categorized into nine food groups by separating leafy and non-leafy vegetables into distinct groups. To address the regional variation of food availability across 8 divisions of Bangladesh, food items exclusively available to regional markets were incorporated into the regional food list while calculating the CoRD of that respective division.

### 2.2. Market Survey on the Price of Food Items

Food prices were collected through a market survey conducted in the last week of January 2021 to 5 February 2021. For collecting price data from eight divisions of Bangladesh, a list of all food markets listed on the Department of Agricultural Marketing (DAM) website was made. The list was then stratified by division (i.e., 8 strata for 8 divisions). Six locations from each division, including 3 urban and 3 rural areas, were randomly chosen to make the final market list comprising 48 markets surveyed.

A two-day training session was conducted to discuss the aims of the market survey and the method of collecting price data for each specific food item. The data enumerators who had previous experience doing market surveys and were familiar with the local language of the assessment area were recruited from each location. The enumerators were provided with pictures of the listed food items to reduce the odds of systematic errors. Before starting the data collection process, formal permission was obtained from the market leaders and local traders to avoid unsolicited circumstances.

The price of food items was collected from four retailers in each market subject to the availability of the food item and retailers in the markets. If any food was not available on the day of data collection, the latest market price of that food item was taken. The 100 g price of every food item from each of the four retailers was recorded, and their average was considered as the food price of that market. The division-specific price of a food item was computed by averaging the prices from 6 locations (e.g., urban and rural areas) of that division. Finally, the average of the prices collected from 48 locations was considered the national-level price of each food item.

It was made sure that the data collectors avoided rush hours, and the prices were collected without causing any disturbance to the traders. As the data collection was conducted amid the pandemic, all our data collectors wore masks and followed hygiene protocols.

### 2.3. Food-Based Dietary Guidelines

In a joint initiative by the Food Planning and Monitoring Unit (FPMU), the Ministry of Food, and the Ministry of Health and Family Welfare of the Government of Bangladesh (GoB), the older 2013 version of FBDG was updated in 2020. This updated FBDG (Figure 2), which provides more quantitative information and specific serving recommendations (Table 1), was used in this study.

The FBDG for Bangladesh is presented in the form of a pyramid of eight food groups such as (1) Cereals; (2) Pulses; (3) Vegetables; (4) Fruits; (5) Meat, fish, and egg; (6) Milk and milk products; (7) Fats and oils; and (8) Sugar. It provides a description of a healthy diet that includes specific serving sizes and the minimum and maximum number of servings from each food group to be eaten in a day for a healthy adult. The FBDG pyramid places leafy and non-leafy vegetables in the same group; however, it instructs that at least one serving of green leafy vegetables should be consumed every day. Thus, we separated leafy and non-leafy vegetables into 2 distinct food groups to make sure we met this condition. Foods under the categories of spices, beverages (except milk), and sweets in the FCTB were excluded while calculating the CoRD, as these are not mentioned in the FBDG of Bangladesh.

### 2.4. Calculating the Cost of Recommended Diet (CoRD)

The calculation of the CoRD consisted of several steps. Firstly, the foods were categorized into specific food groups according to the FBDG. In the case of multiple varieties of the same food, their average price was taken. For example, the average price of *red wheat flour* and *white wheat flour* was taken as the price of *wheat*. The weights of all the food items were standardized into grams. Items that are normally measured in non-standardized units were also converted into grams, such as a dozen eggs and a dozen bananas.

In the second step, the price of the food items as purchased was converted into the price per 100 g edible portion by dividing the “as purchased price” by the edible coefficient. The edible co-efficient value of each food was taken from the FCTB. Next, the price of 100 g of edible food was multiplied by the serving size for each food group recommended in the FBDG of Bangladesh to estimate the price per edible serving.
Price per edible serving=Price per 100 g edible portion of food×Recommended servings of food group

In the third step, we took the average price per edible serving of the 2 lowest-cost items from each food group (see Appendix A) and multiplied it by the average of the upper and lower bound of the number of servings recommended for that group. We chose the lowest-cost items as our objective was to calculate the minimum cost of meeting the recommended diet, and more than 1 lowest-cost item was chosen as the dietary guideline promotes diversity within food groups.

Finally, the costs for meeting the recommendations for each food group were summed to calculate the CoRD.
CoRD=Σ Average price per edible serving of least cost items×Average recommended servings

### 2.5. Measuring Affordability

To assess affordability, the proportion of households in the whole country and in each division that could not afford the CoRD was calculated. Data on household size and daily food expenditure of every household were taken from the 16th round of the HIES (i.e., survey data collected by the Bangladesh Bureau of Statistics from April 2016 to March 2017). Briefly, this survey included 46,080 households from 2304 primary sampling units following a two-stage stratified cluster sampling design. Further details on the planning and implementation of the survey are available elsewhere [26]. As the CoRD was calculated for an adult individual, the reported household size was adjusted with adult male equivalent (AME) values. By multiplying the cost by AME-adjusted household size, the CoRD was determined for every household. As the cost of diets was computed using the food prices of 2021 while the expenditure data were from 2016, the CoRDs were multiplied by a deflation factor. We estimated the deflation factor to be 1.269, 1.271, and 1.264 for national, urban, and rural areas, respectively. Then the deflation-adjusted cost was divided by every household’s daily food expenditure, and the results were expressed in ratios. Ratios above 1 indicated a diet to be unaffordable as the cost exceeded the average food expenditures of a household.
$$Unaffordability\;\left(\%\right)=\frac{Adjusted\;household\;size\times CoRD\times Deflation\;factor}{Household\;daily\;food\;expenditure}$$

## 3. Results

### 3.1. Cost of Recommended Diet

The CoRD was $0.87 per day for an adult person at the national level. The regional variation in the CoRD was clearly observed across the eight divisions, with the highest ($0.93) being in the Sylhet division and the lowest in the Barisal division ($0.64). Recommended diets were more expensive in urban areas for all divisions except Barisal. The residents of Dhaka, the capital and most densely populated region, needed to spend $0.83 on average (Table 2).

### 3.2. Percentage Share of the CoRD and Actual Expenditure

The food expenditure of the Bangladeshi people was dominated mainly by starchy staples rather than protein-rich animal-source foods, dairy products, fruits, and vegetables. According to the expenditure data of HIES, 2016 survey, households spend the lion’s share of their food expenditure on starchy staples (38%), whereas it needs to spend only 21 percent to meet their daily requirements of cereals according to the FBDG. In contrast to staples, households spend only 35 percent of their expenses on protein and 3 percent on dairy products, but to meet the recommended servings, they need to pay 43% and 16%, respectively. On the other hand, they spend 200 percent more on fats and oils compared to what is required for a recommended diet (Figure 3).

### 3.3. Cost Share of Each Food Group in Recommended Diet

Figure 4 represents the cost contribution of each food group according to FBDG in comprising the total cost of a recommended diet. To the total cost, “meat, fish and egg” and “milk and milk products” contribute the major share nationally and in urban and rural areas. This means that to meet the FBDG of Bangladesh, an individual would need to spend more on meat, fish, and egg, along with milk and milk products. The cost of meeting the recommended number of servings of all food groups except leafy vegetables, sugar, and fruits was higher in rural areas than in urban areas (Figure 4, Panel A).

Panel B in Figure 4 presents the percent cost contributions of the food groups across eight divisions where meat, fish, and eggs drive up the cost of a healthy diet in all divisions, especially in Rangpur. Followed by meat, fish, egg, milk and milk products are the food group that costs more in Dhaka, Chattagram, Mymensingh and Sylhet divisions. The cost of meeting recommended amount of cereals is the highest in Barisal and the lowest in Rangpur. To attain the CoRD, one must pay relatively more for fruits and leafy vegetables in Dhaka, whereas an individual from Rangpur would have to pay the least. The results indicate that not only does the cost of the diets vary with geographical location, but also the cost of each food and food group differs simultaneously.

### 3.4. Affordability of the CoRD

As the cost of diets varied with the region and residential area, the affordability of the diets also differed across divisions and areas (Table 3). Nationally about 41.3% (95% CI: 40.8–41.7%) households could not afford the CoRD. The burden of unaffordability was significantly greater in rural (42%) than in urban (39%) areas. The analysis revealed that the highest percentage of households who could not afford recommended diets were from the Khulna division, which was 65.5% (95% CI: 64.5–66.7%), and the fewest in the Chattagram division, which was 25.5% (95% CI: 24.5–26.4%). We also analyzed the district-wise proportion of households unable to afford a recommended diet (see Appendix A).

Likewise, in administrative units and residences, the affordability also altered with geographical locations. Figure 5 shows the percentage of households unable to afford the CoRD in 64 districts of Bangladesh. The percentage of households lacking affordability was divided into five categories ranging from 9.3% to 75.5%. As the percentage of households unable to afford the CoRD increased, the color of the areas got darker. This indicates that the geographical areas with the darkest color were the districts where the freight of unaffordability was the highest. The Choropleth map shows that unaffordability was the highest in the Southwest part, followed by the Northwest part of Bangladesh.

## 4. Discussion

This study aimed to estimate the minimum cost for meeting the FBDG recommended diet (CoRD) of Bangladesh and its affordability. We found that to afford the CoRD, a person must pay $0.87 (83 BDT) per day at the national level, and affordability varied significantly with both residence (e.g., rural and urban) and regions across the eight divisions. The study also showed the percentage an average household spends on individual food groups compared to what they should be paying to meet the FBDG recommended amount. According to the affordability analysis, the prevalence of unaffordability was higher in rural areas compared to urban areas, and the burden of unaffordability was the highest among households in the Khulna division.

The cost of the diets fluctuated largely across regions. These differences may occur likely due to disparities in the prices of foods, especially nutritious foods, as they are often highly perishable and less tradable. A study by Headey and Alderman reported that prices of foods are sensitive to factors such as local food productivity and the efficiency of food value chains [29]. Fluctuations in the cost of diets have also been observed in several previous studies in Bangladesh. For example, to determine the cost of a nutritious diet in the Rangpur division, Save the Children in the year 2007 estimated that the daily cost of meeting a nutritious diet for a family was $0.75 (71 BDT) in the lean season [13]. Later in 2019, the World Food Program (WFP), using the data on food prices and household food expenditure from HIES of 2016, reported the cost of a nutrient-adequate diet was $1.83 (174 BDT) [16]. A more recent study by World Bank and another study by Dizon and Herforth found that to meet the requirements of a healthy diet, and one must pay $0.61 per (58 BDT) day [1,25]. These findings significantly differed from the CoRD we estimated at the national level. Differences in these costs may be attributed to several reasons. One major reason is that this study used the updated FBDG of 2020, which provides more robust information on serving sizes and amounts than the previous FBDG of 2015 and 2013. Moreover, the food prices were collected during the COVID-19 pandemic, which may have influenced the estimates of the CoRD. As per the United Nations report, the prices of food unexpectedly increased during the lockdown, specifically in the rural areas where food supplies had been greatly hampered [30]. Additionally, in this study, the food list used to calculate the CoRD contained 124 food items, whereas the previous studies used a food list from HIES containing only 82 food items.

At the national level, the results showed that 4 out of 10 households were unable to afford the CoRD, and the unaffordability was prominent in the Southwest districts of Bangladesh. Apart from the Southwest part, some districts of the Northwest also had a higher burden of unaffordability compared to the other parts of the country. These variations may attribute to differences in food prices and the purchasing ability of the population. In the report by the World Bank named “Food for improved nutrition in Bangladesh,” the authors stated geographical differences as one of the factors for unaffordability [31]. The report showed that due to the higher diet cost, the people of the Northwest and Southeast parts of Bangladesh spend less on food than the CoRD. Seasonality was another influencing factor, as, during the lean season (July to August), people have less money and face difficulty affording the CoRD than any other period of the year. Unaffordability also varied with the residence as it was higher among the households in rural areas than the urban ones. One reason might be the households of rural areas earn less than those of urban areas and thus exercise less purchasing power. Another reason may be the inadequate storage and processing facilities; the prices of perishable foods are likely to increase in rural areas. Apart from the influence of price, higher unaffordability in rural areas may also be due to the different expenditure patterns of rural households. Often households in rural areas raise consumer crops and livestock for their own consumption, causing them to spend less on foods from markets.

One of the major findings of this study was that among all the food groups, staples possessed the largest share of household expenditure (38%). The households significantly underspent on protein-rich animal-source foods, fruits, leafy vegetables and dairy products, whereas they spent 180% and 200% more on cereals and oils, respectively. The most likely explanation behind this is the drastic rise in the cost of diet when protein-rich animal-source foods (meat, fish, and eggs) and dairy are added to the food basket compared to cereals, pulses, and oil. The implication is that it is costlier to meet the recommendations for dairy and protein-rich animal-source foods, and thus households choose to under-consume them. The high prices of these food groups are also observed in literature of South Asia, where they presented animal foods and dairy products as the costliest food groups and less affordable ones [18,21,25,31]. According to the State of Food Security and Nutrition in the World report (SOFI), low productivity, lack of fair-trade policies, inefficient supply chains, and insufficient local storage capacities are the critical factors to rocketing the prices of nutritious foods [2].

The study results have several implications for making recommended diets less costly and affordable. To reduce the cost of recommended diets, the government should focus on agricultural policies and public food procurement policies to increase the productivity and diversity of foods. In parallel, the government would need to strengthen the market infrastructure and supply chains to allow the flow of diverse nutritious foods into markets, especially dairy, fruits, vegetables and protein-rich animal-source foods. It should also encourage local small and medium entrepreneurs (SMEs) to make investments and innovations to increase the production of indigenous food items to ensure affordable food prices for the poor. A robust nutrition education and behavior change communication (BCC) program through various channels should be undertaken to bring about changes in rice-based food habits and make people aware of nutrient-rich yet relatively cheap food items. To increase affordability, the government should enhance the coverage of existing social protection programs (e.g., open market sales, employment generation programs, and cash for work) to protect the population’s purchasing power.

Though this study gives valuable insights into the cost and affordability of recommended diets, some limitations exist. Firstly, the CoRD method does not consider the local taste and food preferences; rather, it selects the two cheapest foods from each food group. However, this study only estimated the CoRD, which may not fully reflect the food culture and consumption behavior of Bangladeshi consumers. Future works on the CoRD in Bangladesh should take local tastes and food preferences into consideration [22]. For example, a food item, even with the cheapest price, may be discarded when ranking the foods within each food group if that food is known not to be a part of the regular diet of a specific population. Secondly, as we collected price data at a single time, the study could not evaluate the seasonal variations in the cost of the diets. Future studies may employ a longitudinal design to collect price data across the seasons and determine whether significant variability in affordability exists throughout the year. On the other hand, the strength of the study is that it used the actual market price of the foods. Though there are several other sources for food prices, such as the Bangladesh Bureau of Statistics (BBS) and household income and expenditure surveys, the study did not use them due to several shortcomings. For example, the consumer price index (CPI) data of BBS-collected price data for a limited number of food items, and the data were not easily accessible. The household survey data are less frequently collected, often after 5–6 years, and do not give the actual or latest price trend. Thus, using the latest food prices and FBDG of 2020 made the estimate of cost more accurate and reliable in the context of the present time. Future studies may collect food price data on a large scale with more food items at different times of the year to address seasonality. In addition, age and sex-specific dietary guidelines can be designed to assess the cost of meeting recommended diet by age, sex, and reproductive status, rather than focusing on only the adult population.

## 5. Conclusions

The cost of meeting recommended diet remains unaffordable to a large proportion of households in Bangladesh. Due to the higher prices of nutritious foods such as fruits, dairy, and animal foods, the cost of recommended diet rises in all regions. Food price increases compromise diet quality and result in the underconsumption of nutrient-dense foods and the overconsumption of starchy staples. To make diets more affordable, appropriate agricultural policies must be implemented to reduce food price volatility and increase the availability of nutritious foods in the local markets. Food and nutritional assistance can be provided as a part of social security programs in conjunction with behavior change communication to improve access to and consumption of recommended diets.

## Figures and Tables

**Figure 1 foods-12-00790-f001:**
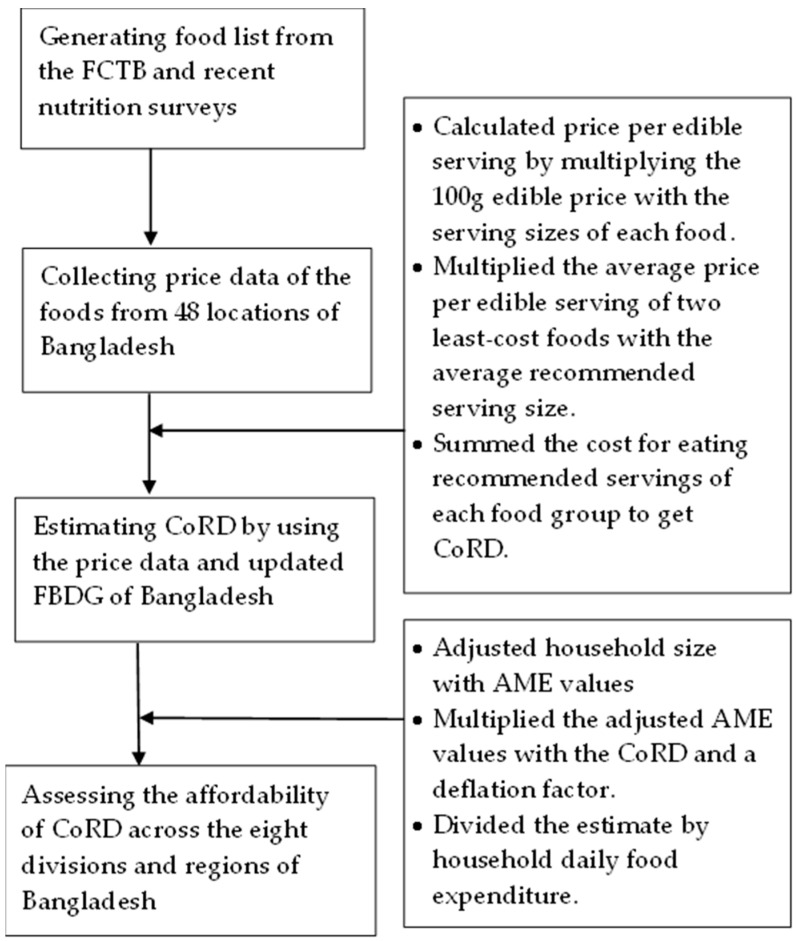
Flowchart of the overall methodology.

**Figure 2 foods-12-00790-f002:**
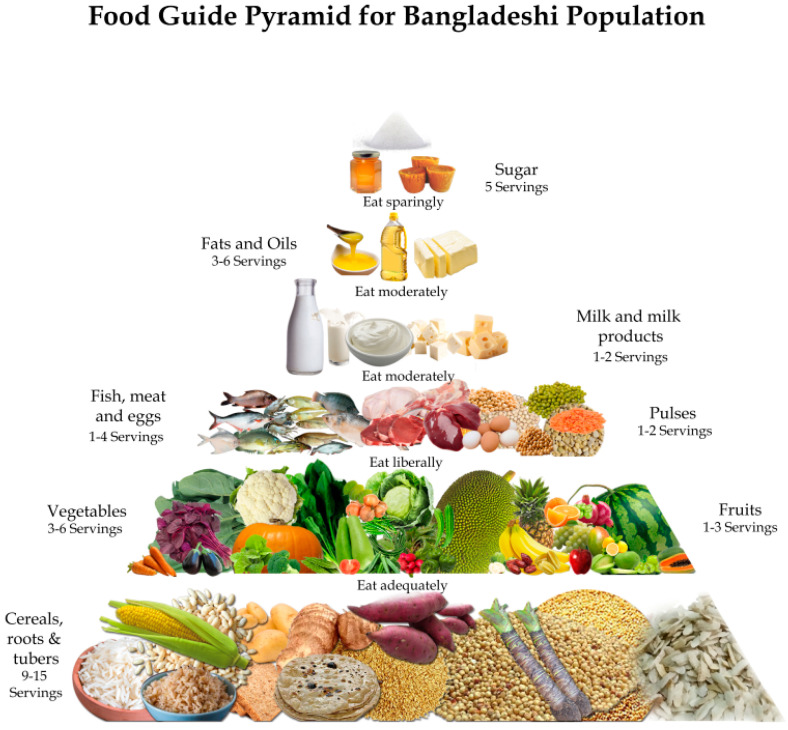
Food-based dietary guidelines of Bangladesh, 2020 (prepared by FPMU, GoB).

**Figure 3 foods-12-00790-f003:**
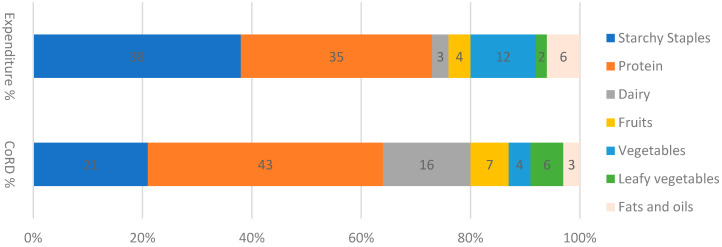
Percentage share of the cost of recommended diet and household food expenditure.

**Figure 4 foods-12-00790-f004:**
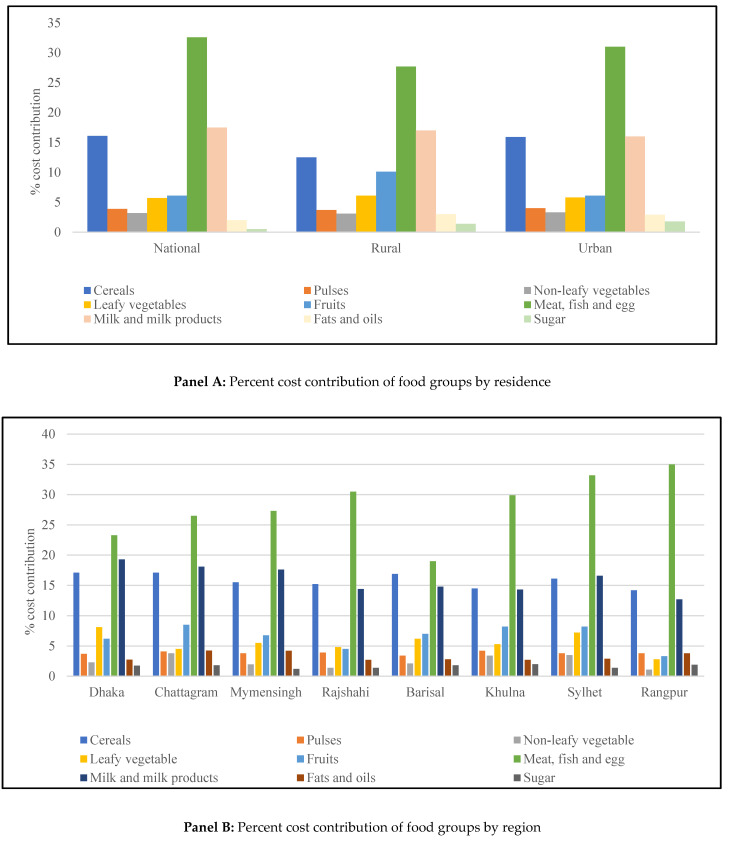
Percent cost contribution of food groups in the recommended diet by residence and region.

**Figure 5 foods-12-00790-f005:**
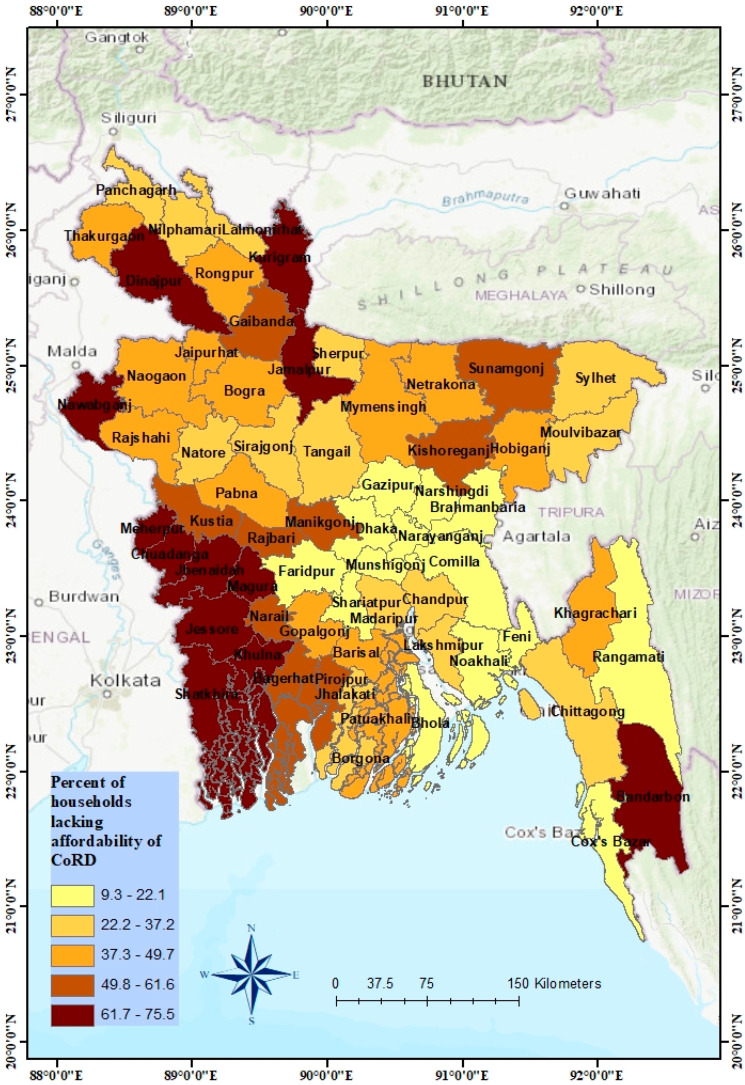
Variation in affordability of CoRD across all districts of Bangladesh.

**Table 1 foods-12-00790-t001:** Serving size estimates based on the food-based dietary guidelines of Bangladesh.

Food Groups		Serving Size (g)	Recommended Number of Servings		Number of Servings Used (Average of the Number of Servings)
			Min	Max	
Cereals		30	9	15	12
Pulses		30	1	2	1.5
Vegetables	Non-leafy vegetables	150	2	4	3
	Leafy vegetables	150	1	2	1.5
Fruits		100	1	3	2.0
Meat, fish, and egg		100	1	4	2.5
Milk and milk products		150	1	2	1.5
Fats and oils		5	3	6	4.5
Sugar		5	5	5	5

**Table 2 foods-12-00790-t002:** The cost of recommended diet (CoRD) in Bangladesh by region and residence.

Locations	Areas	CoRD $ (BDT) * (BDT)
Dhaka	Urban	0.93 (88.4)
	Rural	0.81 (77.3)
	Whole division	0.83 (79.8)
Chattagram	Urban	0.94 (89.2)
	Rural	0.75 (70.6)
	Whole division	0.89 (84.9)
Mymensingh	Urban	0.92 (87.1)
	Rural	0.80 (76.3)
	Whole division	0.85 (80.5)
Barisal	Urban	0.64 (61.3)
	Rural	0.78 (74.5)
	Whole division	0.74 (70.6)
Rajshahi	Urban	0.81 (77.7)
	Rural	0.73 (69.1)
	Whole division	0.79 (75.6)
Khulna	Urban	0.94 (89.5)
	Rural	0.92 (88.0)
	Whole division	0.84 (81.4)
Sylhet	Urban	0.98 (92.5)
	Rural	0.88 (83.3)
	Whole division	0.93 (88.9)
Rangpur	Urban	0.81 (77.6)
	Rural	0.68 (64.1)
	Whole division	0.79 (75.9)
National	Urban	0.90 (85.4)
	Rural	0.84 (80.7)
	Whole country	0.87 (83.0)

* 1$ = 94.7 BDT.

**Table 3 foods-12-00790-t003:** Percent of households unable to afford the recommended diet.

		Unaffordability of CoRD	
		% Households	95% Confidence Interval
By administrative unit	National	41.3	40.8–41.7
	Barisal	35.6	34.1–37.0
	Chattagram	25.5	24.5–26.4
	Dhaka	30.4	29.5–31.4
	Khulna	65.6	64.5–66.7
	Mymensingh	47.3	45.5–49.2
	Rajshahi	45.6	44.4–46.9
	Rangpur	47.5	46.2–48.8
	Sylhet	40.3	38.4–42.1
By residence	Rural	42.5	41.8–42.9
	Urban	39.0	37.9–39.5

## Data Availability

The datasets for this study are available from Bangladesh Bureau of Statistics (BBS) (http://www.bbs.gov.bd/) upon purchase by the researchers.

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
