# Peer review of "Cost of Recommended Diet (CoRD) and Its Affordability in Bangladesh"

_foods, 2023, doi:10.3390/foods12040790_

Round 1
Reviewer 1 Report
This is a paper describing the cost of diet and affordability in Bangladesh. The authors provide valuable information to understand affordability of food in the different regions in the country. I find in general the paper well written and the information valuable. I only have a couple of suggestions to the authors.
1. Please add some description to the method used to collect HIES data. I do understand that these are data from the Bureau of statistics in the country, but I always like to have a brief description of the scope of the survey and a link to further information.
2. I would like to read both a better description of the differences between the COD and the CORD methods, as well as a reflection from the authors on the advantages and weaknesses of each of them. Is there any suggestion to improve CORD after your work?
3. I like the reflection on the policies, but it seems just a list of items without any reference to the context (Bangladesh). Is there anything that could be more or less important in this context?
4. I also like very much the comparison between the actual expenditures and the recommended expenditures (COrd). I was curious to read more about diffences across regions of at least between urban and rural. Are there any potential explanations for the expenditure trends beyond money? are starchy products more readible available? cultural issues?
I hope you find my comments useful, and I wish you luck in your research.
Reviewer 2 Report
Study focuses on estimate the minimum cost for meeting the FBDG recommended diet (CoRD) of Bangladesh and its affordability. The relatively high cost of nutritious foods can affect nutritional outcomes. Price elasticities differ across countries and foods, higher relative prices can generally result in reduced consumption of nutritious foods
In the introduction, it is described that earlier studies assessed FBDG, which is not specific to Bangladesh. What exactly are the differences, besides the old cost of products? It is important to indicate the characteristic differences for the country.
FBDG for Bangladesh (Table 1) - probably worth reporting calories.
Figure 1 shows physical activity, which is not mentioned in the text. How was this factor taken into account?
Were the calorie content of the portions from the lowest-priced foods included in 2.4?
For the applied formulas for calculation, it is worth indicating the literary source.
Rural areas are also likely to raise consumer crops and livestock for their own consumption. How was this factor taken into account?
The materials and methods do not indicate the studied administrative units. Have all regions been taken into account? On fig. 5 shows a large number of regions of Bangladesh, how do they compare with the locations shown in tables 1 and 3? On fig. 5 not all regions are clearly visible. Perhaps, in this figure, the regions under study should be highlighted.
How many households participated in the study from each region?
